# COVID-19 challenges to dentistry in the new pandemic epicenter: Brazil

**Rafael R. Moraes**[1,2]*, **Marcos B. Correa**[1,2], **Ana B. Queiroz**[1], **Ândrea Daneris**[1,2], **João P. Lopes**[1], **Tatiana Pereira-Cenci**[1,2], **Otávio P. D'Avila**[1], **Maximiliano S. Cenci**[1,2], **Giana S. Lima**[1,2], **Flávio F. Demarco**[1]

**1** Federal University of Pelotas, Pelotas, RS, Brazil, **2** GODeC: Global Observatory for Dental Care, Pelotas, RS, Brazil

\* rafael.moraes@ufpel.edu.br

**Data Availability Statement:** The data underlying the results presented in the study are available at https://osf.io/dnbgs/.

## Abstract

A nationwide survey of dentists was carried out in Brazil, a new pandemic epicenter, to analyze how dental care coverage has been affected in public versus private networks, changes in routine and burdens, and how local prevalence of COVID-19 affects dental professionals. Dentists were recruited via email and Instagram®. Responses to a pre-tested questionnaire were collected May 15–24, 2020. COVID-19 case/death counts in the state where respondents work was used to test associations between contextual status and decreases in weekly appointments, fear of contracting COVID-19 at work, and current work status (α = 0.05). Over 10 days, 3,122 responses were received (response rate ~2.1%) from all Brazilian states. Work status was affected for 94%, with less developed regions being more impacted. The pandemic impact on clinical routine was high/very high for 84%, leading to varied changes to clinic infrastructure, personal protective equipment use, and patient screening, as well as increased costs. COVID-19 patients had been seen by 5.3% of respondents; 90% reported fearing contracting COVID-19 at work. Multilevel models showed that greater case and death rates (counted as 1000 cases and 100 deaths per million inhabitants) in one's state increased the odds of being fearful of contracting the disease (18% and 25%). For each additional 1000 cases/100 deaths, the odds of currently not working or treating only emergencies increased by 36% and 58%. The reduction in patients seen weekly was significantly greater in public (38.7±18.6) than in private clinics (22.5±17.8). This study provides early evidence of three major impacts of the pandemic on dentistry: increasing inequalities due to coverage differences between public and private networks; the adoption of new clinical routines, which are associated with an economic burden for dentists; and associations of regional COVID-19 incidence/mortality with fear of contracting the disease at work.

## Introduction

Brazil has emerged as a new COVID-19 pandemic epicenter with steadily growing caseloads. By July of 2020, Brazil was the country with the second-most cases and deaths [1]. With dentistry being a context of high contraction risk and the international supply of personal

**Funding:** This study was financed by Fundação de Amparo à Pesquisa do Estado do Rio Grande do Sul (FAPERGS), Brazil (PRONEX 16/2551-0000471-4, grant recipient FFD), Coordenação de Aperfeiçoamento de Pessoal de Nível Superior (CAPES), Brazil (Finance Code 001, institucional grant) and CAPES, Brazil (PRINT 88881.309861/2018-01, grant recipient FFD). The sponsors had no role in study design, collection, analysis or interpretation of data, writing the report, or decision to submit for publication.

**Competing interests:** The authors declare no conflicts of interest associated with this manuscript.

protective equipment (PPE) compromised, the pandemic has brought major challenges to the dental sector, including maintaining universal dental care coverage for 211 million people dispersed across an 8.5-million-km$^2$ area. Brazil, which has more than half a million dental professionals, including 348,000+ dentists [2], and accounts for an approximately 2.5% share of the 29+ billion USD global market [3], has the most important dental industry in Latin America.

Studies on pandemic effects to dentistry have been carried out in other countries. A survey with 440 dentists in Italy [4] reported that 68% were afraid of being infected during dental procedures during a lockdown period. A survey with 669 dental practitioners from 30 countries [5] showed that 87% were afraid of getting infected with COVID-19 from patients or co-workers, and 90% felt anxious while treating patients who coughed or were suspected of COVID-19 infection. In contrast, a survey with general population in Spain [6] indicated that above 90% of respondents were not afraid of contracting COVID-19 at dental offices, nor would cancel a dental appointment. These differences in behavior could be related to factors including differences in local COVID-19 contraction and death rates, or even how local healthcare systems are managing the pandemic. There is also evidence of changes in clinical routines in dental offices: a survey with 287 dentists in Saudi Arabia [7] showed that 65% of clinics had a workflow for COVID-19 patient screening and new management routines, including body temperature measurements and social distancing in the waiting room. In Italy, telephone triages in dental offices have been reported by 57% of dentists [4], and 80% reported improved training routines on how to wear, remove, and dispose PPE.

While high-quality technological dentistry is available in the private sector in Brazil, low-income citizens depend on a public healthcare system, which showed signs of struggling to cope with the pandemic [8]. Dentistry personnel seem to be facing new routines, more expensive and less comfortable PPE, fewer appointments, and less revenue. These challenges are superimposed upon already existing economic instability in Latin America that has persisted since mid-2014. In this context, dentists may be challenged with fears of contracting COVID-19 while working in a quickly-changing, turbulent situation. Dental teams need to make preventive care efforts to ensure that they do not contribute to worsening the epidemiology of the pandemic. Moreover, the situation could be worsened due to Brazil being in a region of developing countries with entrenched inequalities [9].

Planning medium- and long-term actions to respond to the challenges facing the dental sector related to the COVID-19 pandemic will require establishing a knowledge of baseline parameters, including estimates of key resources, of the sector. In addition, understanding the initial signs of how the pandemic affected the Brazilian dental sector could help other countries in the region to prepare for possible impacts to dentistry. In this study, we conducted a survey with dentists in Brazil, the aims of which were to assess COVID-19 pandemic effects on (i) dental care coverage, (ii) dental office routines and economic burdens, and (iii) the behavior of dentists. The nationwide survey was conducted in May 2020, when the contagion curve was escalating in Brazil.

## Methods

### Study design

The study protocol was approved by the research ethics board from the Medical School, Federal University of Pelotas, Brazil, in May 8, 2020 (#4.015.536). A questionnaire was developed, pre-tested, and used in a cross-sectional open survey with a large sample of dentists in Brazil. The objective of the survey was to address key questions that could impact the dental sector in Brazil as the country was a new pandemic epicenter. The questions were designed to provide

data on possible changes in dental coverage between public and private assistance networks and new dental office routines that could be associated with economic burdens for dentists. In addition, the instrument was designed to assess how dentists were behaving during the pandemic, including their confidence of seeing COVID-19 patients and their fear of contracting the disease at work. The strategy for recruiting participants combined emails sent to dentists and a social media campaign, as further detailed. In order to maximize participation, the questionnaire was designed to be short and having only close-ended questions. In accordance with open science practices, the research protocol, questionnaire in its original language, databank of responses, and other information related to this study are published in an open platform (doi:10.17605/OSF.IO/DNBGS). An English translation of the questionnaire is presented in S1 Table. SURGE guidance [10] and CHERRIES reporting guideline [11] were consulted for this article, which does not cover the full survey content.

## Questionnaire development and pre-testing

A self-administered questionnaire was developed through consultation with eight experienced dental researchers in three discrete review rounds. The questionnaire was hosted online in Google Forms (Google; Mountain View, CA, USA). To obtain information about the reliability and validity of the tool and items, we conducted a pre-test in a sample of 22 dentists who were asked to evaluate its clarity, writing style, question sequence, and internal consistency. The population of pre-testers included differences in sex, age, working sector, region of country, experience, and education levels in an endeavor to resemble the population of dentists in Brazil [2, 12]. The pre-testers were asked to respond the questionnaire and record the time to complete; the mean time to complete ± standard deviation (SD) was 7 ± 2 min.

Pre-testers scored the clarity of each question on a scale of 1 (not clear) to 5 (very clear). A text box was available after every question for pre-testers to explain their scores and place comments, critiques, suggestions, and other response options. All items with a score ≤3 (n = 9) were discussed by at least three researchers to obtain a consensus regarding how to improve them based on pre-tester feedback and then edited accordingly. The mean clarity scores ± SD were high in all cases, varying from 4.79 ± 0.10 for the 9 items that needed revision to 4.91 ± 0.11 for all 30 items considered together. The individual mean score of each question was ≥ 4.86 in 25 questions, and between 4.59 and 4.82 in 5 questions (#5, #9, #12, #14, and #23). The pre-test was important to include other response options in questions #5, #9 and #23, which aided in reducing response bias. Different regulatory authorities were aggregated in question #12, and online training was grouped with general instructions in #14. These groupings were important to avoid overlapping between questions that did not collect multiple answers. Since the changes were minor, the decision was that a second pre-test round was not necessary. The questionnaire was reviewed and revised iteratively by the executive group for approval. Pre-testers were precluded from participating in the main study to avoid response bias.

## Questionnaire content

The first page of the questionnaire contained the study title and objective, an invitation for only dentists to participate and complete the questionnaire only once. They were noticed that their participation was voluntary and not paid, given potential risk and benefit information, and assured that all responses would be treated confidentially and anonymously. In addition, the respondents were asked not to participate if they were not dentists and not to respond the survey again if they have already done it before, reducing the risk for duplicate answers. Participants were directed to print or save the first page of the questionnaire as a PDF file to retain a

copy of the informed consent form. Contact information of the researchers and institution responsible for the survey were provided. The participant had to click 'Yes' after the question "Do you agree to participate in the study voluntarily?" to access the questionnaire. The definitive questionnaire contained 30 mandatory close-ended items (three screens), divided into three sections: demographic and professional profile (n = 8); professional practices during the pandemic (n = 11); and structure and routine of the respondent's main workplace (n = 11). No randomization of items or adaptive questioning were used. The main outcomes were related to the professionals' behavior regarding their clinical routines. The options 'I'd rather not say', 'I don't know how to answer', and 'Does not apply' were available to avoid response errors (see the S1 Table for details about questionnaire content).

## Participant recruitment and survey administration

A source population of 24,126 registered dentists were sent email invitations to participate. The source list was provided in April 2020 by the Brazilian Ministry of Health. The email contained a brief statement that included the study objective, the average response time, notification of the university conducting the study, and a website link to the questionnaire. The initial emails were sent on May 15, 2020; reminder emails were sent 5 days later. Additionally, we created an Instagram social networking campaign targeting dentists in Brazil (Facebook, Menlo Park, CA). To our best knowledge, this is the first study to use Instagram to recruit healthcare professionals. This social network is highly used by dentists in Brazil; as of, July 8, 2020, there were 5.1 million and 6.8 million posts with #dentistry and #odontologia (Portuguese for dentistry). The campaign, which started on May 20, invited dentists to participate in an online survey regarding the impact of the pandemic on their practices. A significant challenge was the fact that Instagram does not allow placing linking in comments or pots. Thus, an Instagram professional account was created (@odcovid) with a website link to the questionnaire in its bio page. Invitations were posted calling for the participation of dentists; they included the same information provided in the email invites and directed the dentists to use the hyperlink available on the @odcovid bio page. We used hashtags related to dentistry and COVID-19 to increase reach to the target population. Participating researchers shared the invitations on their personal Instagram profiles (feed and stories) and asked other dentists to aid in disseminating the campaign. Brazilian dentists with professional Instagram profiles were asked to also share the invitation post. We reached professionals categorized as micro (<10,000 followers) and meso (10,000–1 million) on the followers scale [13]. A second Instagram campaign with similar content but a slightly different visual presentation was created two days later.

## Sample selection, sample size estimation, and collection of responses

All dentists practicing in Brazil were eligible. Given a target population of ~348,000 professionals, we estimated that 2,385 responses would be necessary to ensure a 95% confidence interval and 2% margin of error. Responses were collected between May 15 and May 24, 2020.

## Data analysis

Partial questionnaire completion was not possible. In some cases, responses were restricted to a specific population. The response options 'I'd rather not say', 'I don't know how to answer', and 'Does not apply' were treated as missing data. Descriptive statistics were used to identify frequencies and distributions of variables. Responses to questions on numbers of patients assisted weekly, before and after the pandemic, were subjected to t-test. Proportions were compared using chi-square tests. COVID-19 case and death counts in each Brazilian state were obtained from official Ministry of Health reports [14] on May 20, 2020, the date when the

greatest number of survey responses was received. For analysis purposes, data were converted into thousands of cases and hundreds of deaths per one million inhabitants in each state. The units of analysis of both variables were the states. Multilevel mixed effect models were used to test the association between the contextual status of the pandemic in each state and dentistry-related outcomes. Outcomes included decrease in number of patients assisted weekly (numerical), fear of contracting COVID-19 at work (no/a little vs. yes/a lot), and current work status (normal/reduced vs. not working/emergencies only). Linear and logistic models were used for numeric and binary outcomes. The models considered two levels of organization: dentist (level 1) and state (level 2). β-coefficients and Odds Ratios (OR) were reported. Contextual level variance was assessed using intraclass correlation coefficient (ICC) for linear models and Median Odds Ratios (MOR) for logistic models (α = 0.05). All analyses were performed in Stata 14.2 (StataCorp, College Station, TX).

## Results

A total of 3,122 valid responses were received over 10 days from all 26 Brazilian states and the federal district. No questionnaires were submitted with an atypical timestamp. Gathering of responses over time is shown in Fig 1A. The first 5 days included only email invitation responses. The response rate in this period was 2.1%, but the numbers of actual rejections/losses cannot be calculated because we cannot estimate how many dentists actually received the questionnaire and decided not to respond, for instance. We received 1,572 responses in the first 24 h after the Instagram campaign started. As shown in Table 1, respondents were most female (75%) and in practice for ≤20 years (74%). Meanwhile, 53% were working mainly in private clinics, whereas 36% were working in the public sector. The mean age ± SD of the respondents was 38 ± 11 years. Table 2 shows a demographic comparison between overall dentists working in Brazil and respondents who participated in the present survey. The distributions of responses by region, sex, and age were similar to the overall distributions of dentists in Brazil, except for a higher response rate from females and from Southern Brazil.

### Dental care coverage

Current work status was reported to be affected by 94% of the respondents. Only 2% reported normal or increased patient volumes. Not working/emergency only statuses were more common among dentists working in the less developed North and Northeast regions, and also in the Southeast region (Fig 1B). Interestingly, 59% of respondents reported be willing to assist or having already assisted patients online, and 26% regarded such virtual consults as being positive experiences.

The proportion of dentists who reported not seeing patients at all due to the pandemic was similar between public and private networks (Fig 1C). However, whereas only 52% of private dentists reported seeing less patients than usual due to the pandemic, 76% of public clinic dentists reported maintaining only emergency appointments, yielding a significant difference on the effect of the pandemic on the volume of patients treated weekly (Table 3). Before the pandemic, the public network covered more patients per dentist. During the pandemic, reductions in weekly dental care levels were reported to be 23 patients/private dentist and 39 patients/dentist in the public network.

The effects of COVID-19 confirmed-case and death rates on the numbers of patients assisted (Table 4) showed dentists seeing two fewer patients/week for each 1000 cases per one million inhabitants, and three fewer patients/week for each 100 deaths. This effect was more pronounced in the public network: 2.45 and 3.25 fewer patients were seen each week for every

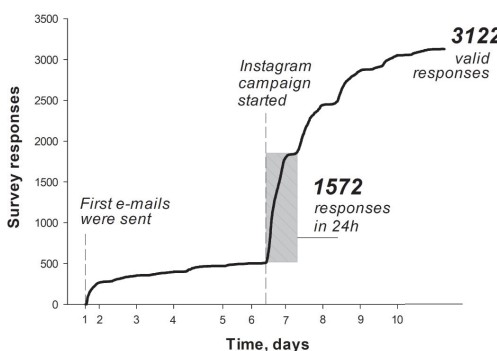

**A: responses gathered with time**

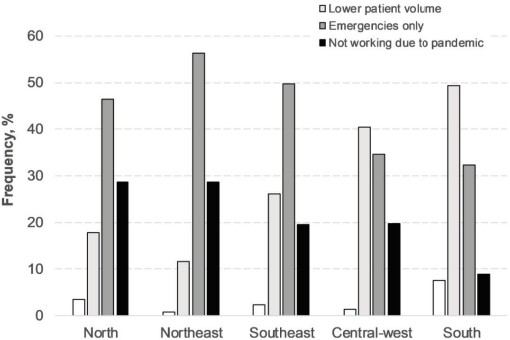

**B: Current work status by region**

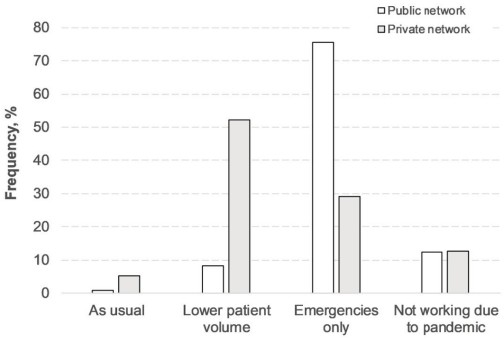

**C: Current work status by sector**

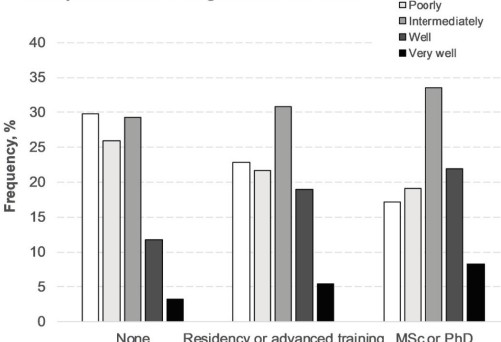

**D: Preparation level vs. graduate education**

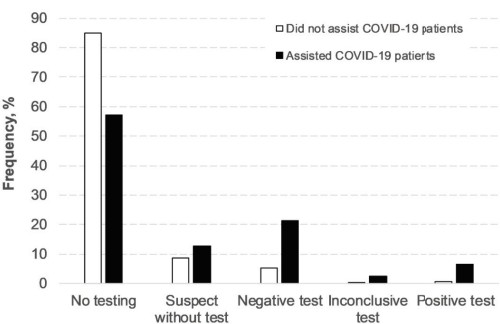

**E: assisting COVID-19 patients vs. testing**

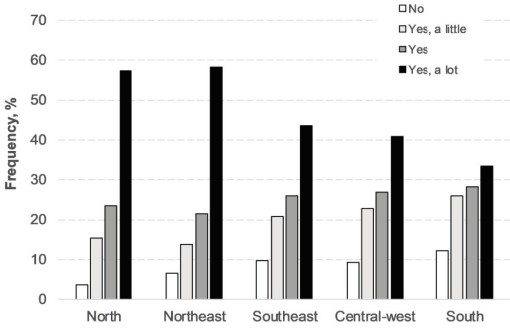

**F: Fear to contract COVID-19 at work by region**

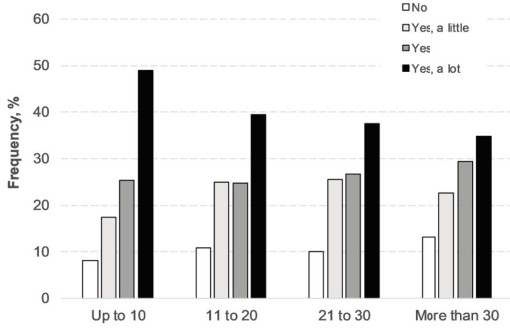

**G: Fear to contract COVID-19 vs. years in practice**

**Fig 1. Factors influencing COVID-19 pandemic effects on dental practices.** (A) Over 10 days, 3,122 valid survey responses were received from all regions in Brazil. (B) The work statuses of 'not working' or 'emergency only' were more frequent in the less developed North and Northeast regions, and also in the Southeast (p<0.001). (C) Work status by sector: 52% of private dentists reported seeing less patients than usual, while most public dentists reported emergency appointments only (p<0.001). (D) Education level influenced how prepared professionals feel to assist COVID-19 patients (p<0.001). (E) Dentists who had confirmed contraction of COVID-19 themselves (6.4%) were more likely (p<0.001) to have assisted patients with COVID-19 (tested positive) than dentists who had not (0.7%). (F) Fear of contracting COVID-19 at work varied across regions, being higher in the North and Northeast regions than in other regions (p<0.001). (G) Fear of contracting COVID-19 at work was influenced by years in practice (p<0.001).

1000 cases or 100 deaths per one million inhabitants, respectively. In this analysis, the number of patients seen by dentists working in private practice was not significant.

## Routine and economic burden for dentists

The impact of the pandemic on clinic routines was considered high or very high by 84% of respondents (0.6% reported no impact). Though 80% of respondents reported increased financial costs, only 15% adjusted prices for patients. The pandemic required infrastructural changes in the work setting for 74% of dentists. Most had new types of PPE available for all clinical appointments, including face shields (84%), N95 masks (71%), and disposable coats (66%). Patient screening became more expensive and time consuming due to antimicrobial mouthwashes (46%), completion of COVID-19 questionnaires (35%), and temperature monitoring (24%) mainly. For 35% of respondents, N95 masks were the predominant mask used (with at least half of patients).

## Behavior of dentists during pandemic

As reported in Table 1, more than four out of five dentists reported undergoing at least some training in COVID-19 preventive measures, though fewer than one in twenty participated in practical in-clinic training. While almost a quarter of respondents reported feeling well/very well prepared to treat patients with COVID-19, only 5.3% had done so (Table 1). Perception of preparedness to provide care for COVID-19 patients was influenced by education level (Fig 1D). It was more common for dentists who treated patients with COVID-19 to also have COVID-19 (6.4%), than for those who had not seen COVID-19 patients (0.7%) (Fig 1E). Testing was also more frequent for dentists who had seen COVID-19 patients. Although 90% feared contracting the disease at work, only 8% indicated that they had been tested for COVID-19 (1.1% had a positive test). Fear varied among regions, being particularly elevated in the North and Northeast (Fig 1F), and with years in practice (Fig 1G). Fear of contracting COVID-19 at work related positively to the numbers of cases and deaths reported in the state in which the respondent was working. Each 1000 cases per million inhabitants and each 100 deaths per million inhabitants increased the odds of having fear to contract COVID-19 (Table 5). Likewise, MOR indicated that, compared to dentists in less impacted states, dentists practicing in more highly impacted states had a more than 30% greater likelihood of fearing that they may contract COVID-19 and were more than twice as likely to be offering emergency only appointments or to be closed altogether rather than maintaining a usual or even reduced volume of patients with full-service availability. For each 1000 cases and each 100 deaths per million residents in the state, the likelihood of not working or treating emergencies only, as opposed to working with a reduced or typical patient volume, increased by 36% and 58%.

## Discussion

Here, we report the findings of the first survey, to the best of our knowledge, in which both email and Instagram social networking campaigns were used to reach healthcare professionals.

**Table 1. Demographic and work practice characteristics of the respondents, Brazil, 2020 (N = 3,122).**

| Variable/category | n* | % | 95% CI |
|---|---|---|---|
| **Sex** | 3,116 | | |
| Male | 790 | 25.4 | 23.9; 26.2 |
| Female | 2,326 | 74.7 | 73.1; 76.2 |
| **Years in practice** | 3,121 | | |
| ≤10 | 1,496 | 47.9 | 46.2; 49.7 |
| 11–20 | 812 | 26.0 | 24.5; 27.6 |
| 21–30 | 501 | 16.1 | 14.8; 17.4 |
| >30 | 312 | 10.0 | 9.0; 11.1 |
| **Postgraduate education (completed)** | 3,121 | | |
| None | 758 | 24.3 | 22.8; 25.8 |
| Residency or advanced special training | 1,530 | 49.0 | 47.3; 50.8 |
| MSc or PhD | 833 | 26.7 | 25.2; 28.3 |
| **Main work sector** | 3,051 | | |
| Public | 1,091 | 35.8 | 34.1; 37.5 |
| Private | 1,601 | 52.5 | 50.7; 54.2 |
| Other | 359 | 11.8 | 10.7; 13.0 |
| **Brazilian regional division** | 3,122 | | |
| South | 1,183 | 37.8 | 36.2; 39.6 |
| Southeast | 923 | 29.6 | 28.0; 31.2 |
| Central-west | 221 | 7.1 | 6.2; 8.0 |
| Northeast | 682 | 21.9 | 20.4; 23.3 |
| North | 113 | 3.6 | 3.0; 4.3 |
| **Current work status** | 3,056 | | |
| As usual | 119 | 3.9 | 3.3; 4.6 |
| Lower patient volume | 994 | 32.5 | 30.9; 34.2 |
| Emergency appointments only | 1,325 | 43.4 | 41.6; 45.1 |
| Not working due to pandemic | 546 | 17.9 | 16.5; 19.3 |
| Not working due to other reasons | 72 | 2.4 | 1.9; 3.0 |
| **Volume of weekly patients compared with pre-pandemic period** | 2,812 | | |
| Increased or normal | 62 | 2.2 | 1.7; 2.8 |
| Reduced | 2750 | 97.8 | 97.2; 98.3 |
| **Have you had online patient appointments during the pandemic?** | 2,832 | | |
| No but I am willing to do | 755 | 26.7 | 25.1; 28.3 |
| No and I am not willing to do | 1,159 | 40.9 | 39.1; 42.7 |
| Yes, the overall experience was positive | 726 | 25.6 | 24.1; 27.3 |
| Yes, the overall experience was negative | 192 | 6.8 | 5.9; 7.8 |
| **Impact of pandemic in work routine** | 3,048 | | |
| No impact | 17 | 0.6 | 0.3; 0.9 |
| Low | 99 | 3.3 | 2.7; 3.9 |
| Intermediate | 389 | 12.8 | 11.6; 14.0 |
| High | 926 | 30.4 | 28.8; 32.0 |
| Very high | 1,617 | 53.1 | 51.3; 54.8 |
| **Have work routine changes led to increased financial costs?** | 2,207 | | |
| No | 447 | 20.3 | 18.6; 22.0 |
| Yes, but prices were not adjusted | 1,432 | 64.9 | 62.9; 66.9 |
| Yes, and prices were adjusted for patients | 328 | 14.9 | 13.4; 16.4 |
| **Training for COVID-19 specific preventive measures** | 3,099 | | |

(*Continued*)

**Table 1.** (Continued)

| Variable/category | n* | % | 95% CI |
|---|---|---|---|
| None | 559 | 18.0 | 16.7; 19.4 |
| Online training or general instructions | 2,406 | 77.6 | 76.1; 79.1 |
| Practical training | 134 | 4.3 | 3.7; 5.1 |
| **Have you treated patients with a confirmed COVID-19 diagnosis?** | 2,401 | | |
| No or do not know | 2,275 | 94.8 | 93.8; 95.6 |
| Yes | 126 | 5.3 | 4.4; 6.2 |
| **How prepared do you feel to treat patients with COVID-19?** | 3,040 | | |
| Not at all prepared | 702 | 23.1 | 21.6; 24.6 |
| Poorly prepared | 670 | 22.0 | 20.6; 23.5 |
| Intermediately | 948 | 31.2 | 29.6; 32.9 |
| Well prepared | 547 | 18.0 | 16.7; 19.4 |
| Very well prepared | 173 | 5.7 | 4.9; 6.6 |
| **Do you fear to contract COVID-19 at work?** | 3,024 | | |
| No | 295 | 9.7 | 8.7; 10.9 |
| Yes, a little | 643 | 21.3 | 19.8; 22.8 |
| Yes | 781 | 25.8 | 24.3; 27.4 |
| Yes, a lot | 1,305 | 43.2 | 41.4; 44.9 |
| **Have you suspected or tested yourself for COVID-19?** | 3,093 | | |
| No | 2,517 | 81.4 | 80.0; 82.7 |
| Suspect without test | 314 | 10.2 | 9.1; 11.3 |
| Negative test | 213 | 6.7 | 6.0; 7.8 |
| Inconclusive test | 16 | 0.5 | 0.3; 0.8 |
| Positive test | 33 | 1.1 | 0.7; 1.5 |
| **Do you agree with current social distancing measures in your city?** | 3,104 | | |
| Fully disagree | 63 | 2.0 | 1.6; 2.6 |
| Partially disagree | 330 | 10.6 | 9.6; 11.8 |
| Not agree or disagree | 38 | 1.2 | 0.8; 1.7 |
| Partially agree | 1,001 | 32.3 | 30.6; 33.9 |
| Fully agree | 1,672 | 53.9 | 52.1; 55.6 |

* Varies from total N because of missing data for different questions. CI: confidence interval.

Details on the populations participating in the survey that were recruited by different approaches were addressed in a separate report [15]. Although the use of social media in research has been discussed [16–18], there is scarce information regarding its use to recruit hard-to-reach populations [19–21]. A combined strategy was important to recruit dentists working in both public and private networks, and doing so allowed us gather one of the largest samples to date for a COVID-19 survey in the dental field [4, 5, 7, 22–26]. The pandemic may have facilitated our recruitment owing to people spending more time at home and on social media [27]. Online surveying methods are particularly important during this time when sanitary measures prevent traditional research approaches [15].

This report in concentrated on the early COVID-19 impacts to the dental sector in Brazil. The present results provide early evidence of three major aspects being at stake in dentistry in the new pandemic epicenter. First, differences in care coverage between public and private clinics suggest an intensification of regional and socioeconomic inequalities. Second, although dentists have a similar fear of contracting COVID-19 at work as other healthcare providers, they seem to report feeling less prepared to assist patients [28]. Third, dentists have adopted

**Table 2. Distribution of dentists working in Brazil by sex, age, and region (%) compared with the survey participants, Brazil, 2020 (N = 3,122).**

|  | Dentists working in Brazil[*] | Survey respondents |
|---|---|---|
| **Sex** |  |  |
| Male | 43.9 | 25.4 |
| Female | 56.1 | 74.7 |
| **Age** |  |  |
| ≤30 | 25.2 | 32.7 |
| 31–40 | 32.2 | 33.4 |
| 41–50 | 23.6 | 20.4 |
| 51–60 | 14.1 | 9.9 |
| >60 | 4.9 | 3.5 |
| **Brazilian region** |  |  |
| South | 16.1 | 37.8 |
| Southeast | 52.8 | 29.6 |
| Central-west | 8.8 | 7.1 |
| Northeast | 16.6 | 21.9 |
| North | 5.7 | 3.6 |

[*]Data obtained from official reports [2, 12].

new routines and incurred increased costs, which eventually will be transferred to patients in the private network, or paid by the government in public clinics. The scenario is aggravated by disjointed responses from the Brazilian government and the associated lack of an effective coordinated national response to the pandemic [29]. The multi-level analysis showed that mounting COVID-19 case and death counts are affecting dentists' behavior in the new pandemic epicenter. Other studies also have observed that the pandemic is bringing fear to dentists at work. In a survey [5], 85% of respondents reported feeling afraid when heard news about COVID-19 caused deaths. In addition, 92% were afraid of carrying the infection from offices to their families. The aerosolized cloud in dental offices is a constant reminder of danger. Training in preventive measures and the use of up-to-date screening methods may be appropriate first steps for dentists to feel better prepared to attend to COVID-19 patients. Individual cognizance and knowledge of pertinent information are important factors in healthcare workers feeling confidence in dealing with and overcoming the pandemic [28].

Brazilian dental sector stakeholders seem to be paying diligent attention to the ongoing situation. Dental councils and sanitary agencies have already released important guidance documents in the meantime. The vast majority of our study respondents (91%) indicated that they are following official regulatory standards in their new routines, and that they, by and large,

**Table 3. Mean numbers of patients treated weekly per dentist by work sector (standard deviation), before and during the pandemic, Brazil, 2020 (n = 2,534 dentists[*]).**

|  | Public network | Private practice | Total |
|---|---|---|---|
| **Before pandemic** | 47.3 (19.7) | 34.2 (20.8) | 39.6 (21.3) |
| **During pandemic** | 8.6 (8.6) | 11.7 (13.6) | 10.2 (11.8) |
| **Difference**[**] | 38.7 (18.6) | 22.5 (17.8) | 29.2 (19.8) |

[*]Varies from total N because of missing data for different questions.

[**]t-test (p<0.001).

**Table 4. Effect of numbers of confirmed COVID-19 cases and deaths* on differences in numbers of patients seen by work sector, Brazil, 2020 (n = 2,534 dentists**).**

| | Effects on decreases in numbers of patients seen | | | | |
|---|---|---|---|---|---|
| **Overall** | **β** | **95% CI** | **P-value** | **ICC[1]** | **ICC[2]** |
| **1000 cases/million inhabitants** | 1.96 | 0.43; 3.49 | 0.012 | 0.101 | 0.086 |
| **100 deaths/million inhabitants** | 2.90 | 0.80; 5.00 | 0.007 | 0.101 | 0.085 |
| **Public network** | | | | | |
| **1000 cases/million inhabitants** | 2.45 | 0.55; 4.36 | 0.012 | 0.151 | 0.144 |
| **100 deaths/million inhabitants** | 3.25 | 0.98; 5.52 | 0.005 | 0.151 | 0.137 |
| **Private practice** | | | | | |
| **1000 cases/million inhabitants** | 1.12 | -1.55; 3.78 | 0.410 | 0.151 | 0.144 |
| **100 deaths/million inhabitants** | 2.34 | -0.95; 5.62 | 0.163 | 0.151 | 0.137 |

*Multilevel linear regression model considering all 26 different Brazilian states and the federal district. CI, Confidence Interval; ICC, Intraclass Correlation Coefficient

[1]Null model

[2]Adjusted model.

**Varies from total N because of missing data for different questions.

have made substantial efforts to cope with the new clinical requirements. In corroboration, studies in other countries [5, 30] addressed that the fear of dentists in getting infected from COVID 19 could be mitigated by dentists and the dental team carefully following recommendations from regulatory authorities. While patient appointment volumes reported in May were significantly below from pre-pandemic levels, our data indicate that Brazilian dentists are open to the incorporation of telehealth programs, which may, despite its associated challenges, be a good strategy for mitigating the impact of the pandemic, while improving preventive actions and reducing unnecessary referrals [31]. A survey in Italy [4] reported that only 37% of dentists considered telehealth as a valid program, and less than 13% reported to have used it before. In contrast, 46% of dental practitioners believed that digital dentistry could be used more often in the post-pandemic period. A recent article [32] recommended the implementation of fully digital approaches during the COVID-19 pandemic to limit infection risks, whenever possible. Another report suggested that although the pandemic has caused many difficulties for provision of dental care, an opportunity is established for dental educators to modernize their teaching approaches using novel digital and online communication [33].

**Table 5. Effect of numbers of confirmed COVID-19 cases and deaths* on fear of contracting COVID-19 at work and current work status, Brazil, 2020 (n = 3,021 dentists**).**

| Variable | OR | 95% CI | P-value | MOR[1] | MOR[2] |
|---|---|---|---|---|---|
| **Fear to contract COVID-19 at work (ref: none/a little)** | | | | | |
| **1000 cases/million inhabitants** | 1.18 | 1.01; 1.39 | 0.039 | 1.42 | 1.32 |
| **100 deaths/million inhabitants** | 1.25 | 1.02; 1.52 | 0.029 | 1.42 | 1.34 |
| **Work status (not working/only urgencies vs. normal/reduced frequency)** | | | | | |
| **1000 cases/million inhabitants** | 1.36 | 1.00; 1.86 | 0.050 | 2.50 | 2.28 |
| **100 deaths/million inhabitants** | 1.58 | 1.06; 2.38 | 0.026 | 2.50 | 2.22 |

*Multilevel logistic regression model considering all 26 different Brazilian states and the federal district. CI, Confidence Interval; OR, Odds Ratio; MOR, Median Odds Ratio

[1]Null model

[2]Adjusted model.

**Varies from total N because of missing data for different questions.

The low volume of patients reported being seen in the public network in Brazil may reflect a prioritization of PPE supplies for healthcare professionals providing medical treatment to COVID-19 patients as well as Ministry of Health directives to provide care for dental emergencies only. Another study showed similar findings in Spain [30], indicating a significant decrease in the number of patients seen by dentists during the national state of alarm; 86% of participants reported seeing up to five patients per week due to recommendations to treat only urgent situations. In the early stages of the pandemic worldwide, a shortage of PPE was also a significant concern. Lack or inadequate availability of PPE has been reported to potentially impose negative impact on the mental health of professionals [34], and may worsen the scenario. One could argue that pandemic-associated increases in the need for medical devices and PPE, and the emerging vaccine industry, should be favorable to business in the biomedical industry. However, in Brazil, this industry accounts for less than 43% of the national consumption production in general biomedical supplies [35]. KaVo, a major dental company worldwide, recently closed its manufacturing facilities in Brazil, which may be an early sign of employment loss in the sector. Government-aided measures to support PPE supply and biomedical industries could be necessary in the long term.

The present study also points out concerns with regard to economic burden associated with changes in routine dental practices. In corroboration, a prior economic analysis showed that COVID-19 mitigation/suppression measures will cause financial distress to private dental clinics in Germany [36]. A total 35% dentists participating in the present survey, for instance, reported that N95 masks were the predominant mask used (with at least half of patients). Taking into account the pre-pandemic volume of patients treated weekly by this sample (average 39.6 patients/dentist), and the ~348,000 dentists registered in Brazil, generalizing the figure of 35% of dentists treating at least half of their patients wearing N95 masks without re-use, dentists in Brazil can be expected to use some 9.6 million masks per month. Considering typical prices for surgical masks (0.4 USD) and N95 masks (2.92 USD) (quotes retrieved by authors and available in doi:10.17605/OSF.IO/DNBGS), the yearly cost of this simple PPE change would be ~290 million USD, which would amount to approximately 1.16 billion USD over a potential 4-year COVID-19 resurgence risk period [37].

It should be noted that our study design does not allow one to establish cause-effect relationships, thus our findings should be treated with caution. In addition, it should be considered that regional COVID-19 rates could be influenced by socioeconomic factors, which may additionally play a role on the behavior of dental professionals. Although the representativeness of the sample may have limitations, the distributions of responses by region, sex, and age were similar to the overall distributions of dentists in Brazil (Table 2), except for higher response rates from females and from dentists working in Southern Brazil. Notwithstanding, our sample variability was supported by the large numbers of responses received. A report from 2009 [12] showed that the proportion of female dentists in Brazil was in a rise, which also may account for the differences in sex observed between respondents here. The present findings also suggest that female dentists could be more likely to engage in survey research than male dentists, at least during the pandemic. Collecting responses by an open campaign on social media also imposes limitations [15], including a higher chance of recruiting dentists who were more afraid about the pandemic or more willing to cooperate with sanitary measures. The fact that the social media campaign was originated in Southern Brazil explains the higher proportion of respondents from the South region [15]. It is also worth mentioning that the South and Southeast regions, which have similar human development index and per capita income values [38, 39], represent the highest-income regions of Brazil.

Future studies will be necessary to monitor how dentists are coping with the pandemic. Data from this study may be useful as a baseline relative to future developments and useful in

designing interventions. The scenario observed in May 2020 can be used for future observations on the impact and evolution of the pandemic to the dental sector in Brazil and Latin America, as well as potentially associated crises. Brazil is a big player in dentistry worldwide, with a particularly predominant role in Latin America. Unfortunately, given its concentrated effects in the public dental care sector, the pandemic appears to be contributing to a deepening of already marked inequalities in oral health within Brazil, and such effects may extend more broadly into Latin America. Inadequate public healthcare funding in the short to long term may increase the risk of exacerbating historical inequalities between regions in Brazil. Actions taken now will affect how Brazilian dentistry is regarded after the pandemic, and whether Brazil will be a good or bad example of dental practices, especially for neighboring countries. Even after the contagion curve is flattened, we can expect precautionary changes to dental clinic routines and associated stress to persist for years given that dental professionals will continue to be at high risk of exposure, especially in the event of a future resurgence. Ultimately, the outlook of the dental sector depends on political, professional, and personal actions in this turbulent period during which major aspects are at stake.

## Conclusions

This survey gathered 3,122 responses from all regions of Brazil in May 2020, when the country was a new pandemic epicenter and the contagion curve was escalating. The results provide early evidence of three major impacts of the pandemic on the dental sector: increasing inequalities due to care coverage differences between the public and private networks; the adoption of new clinical routines, which are associated with an economic burden for dentists; and associations of regional COVID-19 incidence/mortality with fear of contracting the disease at work. Constant monitoring of the situation is encouraged over the course of events in the ongoing pandemic.

## Supporting information

**S1 Table. Questionnaire (original language: Brazilian Portuguese).**
(DOCX)

## Acknowledgments

We thank the Brazilian Ministry of Health for providing the source list of professionals. We also thank all persons who helped to disseminate the campaign on Instagram and dentists who volunteered to participate.

## Author Contributions

**Conceptualization:** Rafael R. Moraes, Marcos B. Correa, Tatiana Pereira-Cenci, Otávio P. D'Avila, Maximiliano S. Cenci, Giana S. Lima, Flávio F. Demarco.

**Data curation:** Rafael R. Moraes, Marcos B. Correa, Ana B. Queiroz, Ândrea Daneris, João P. Lopes, Giana S. Lima.

**Formal analysis:** Marcos B. Correa.

**Funding acquisition:** Flávio F. Demarco.

**Investigation:** Rafael R. Moraes, Marcos B. Correa.

**Methodology:** Rafael R. Moraes, Marcos B. Correa, Ana B. Queiroz, Ândrea Daneris, João P. Lopes, Tatiana Pereira-Cenci, Otávio P. D'Avila, Maximiliano S. Cenci, Giana S. Lima.

**Project administration:** Rafael R. Moraes.

**Resources:** Rafael R. Moraes.

**Software:** Marcos B. Correa.

**Supervision:** Rafael R. Moraes, Flávio F. Demarco.

**Validation:** Rafael R. Moraes, Marcos B. Correa, Ana B. Queiroz, Ândrea Daneris, João P. Lopes, Maximiliano S. Cenci.

**Writing – original draft:** Rafael R. Moraes, Marcos B. Correa.

**Writing – review & editing:** Rafael R. Moraes, Marcos B. Correa, Ana B. Queiroz, Ândrea Daneris, João P. Lopes, Tatiana Pereira-Cenci, Otávio P. D'Avila, Maximiliano S. Cenci, Giana S. Lima, Flávio F. Demarco.

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
