## [Decision Letter · Decision Letter 0]

20 Aug 2020

PONE-D-20-21705

COVID-19 challenges to dentistry in the new pandemic epicenter: Brazil

PLOS ONE

Dear Dr. Moraes,

Thank you for submitting your manuscript to PLOS ONE. After careful consideration, we feel that it has merit but does not fully meet PLOS ONE’s publication criteria as it currently stands. Therefore, we invite you to submit a revised version of the manuscript that addresses the points raised during the review process.

We look forward to receiving your revised manuscript.

Kind regards,

Srinivas Goli, Ph.D.

Academic Editor

PLOS ONE

Journal Requirements:

2. Thank you for including your ethics statement:  "The study protocol was approved by our institutional research ethics board (#4.015.536) in May 8, 2020"

Additional Editor Comments (if provided):

Although manuscript is suitable for publication but it is poorly written. Authors must improve the presentation of the paper in line with reviewers comments.

Reviewers' comments:

Reviewer's Responses to Questions

**Comments to the Author**

1. Is the manuscript technically sound, and do the data support the conclusions?

Reviewer #1: Partly

Reviewer #2: Partly

2. Has the statistical analysis been performed appropriately and rigorously? 

Reviewer #1: Yes

Reviewer #2: Yes

3. Have the authors made all data underlying the findings in their manuscript fully available?

Reviewer #1: Yes

Reviewer #2: Yes

4. Is the manuscript presented in an intelligible fashion and written in standard English?

Reviewer #1: Yes

Reviewer #2: Yes

5. Review Comments to the Author

Reviewer #1: This study aimed to assess COVID-19 pandemic effects on dental coverage, dental office routines and economic burdens, and the behaviour of dentists.

1. The sampling methodology is poorly described, for example, how do you know that a dentists (anonymously) belong to each state?

2. Covariates criteria and categories are simple and unclear, such as: did both the 1000 cases /100 deaths per million inhabitants in each state represent each state or a unit?

3. Data analysis are also simple and unclear, especially for multilevel models: what was ‘null model’? and what is difference ‘OR’ and “MOR’? were models adjusted for any covariates?

4. There is insufficient discussion of the relevance and the aim of the study following the findings/results.

5. Discussion section: it is short of strength and limitation reporting.

Reviewer #2: 1. Introduction doesn't cover enough background of research conducted so far on the topic.

2. Methods section is too lengthy and has many repeated explanations

3. The response rate is very low though the sample size is adequate. since its a national wide survey, the response rate is critical. so i would suggest to improve the response rate and subsequent analysis

4. Discussion is too general and not based on important findings of the study. I would suggest to compare findings with other countries. It is more important to discuss on how these findings helps for change in policy as mentioned in the manuscript. further discuss the impact of pandemic on dentistry in other countries.

5. Please check the references as some are repeated in the list.

6. PLOS authors have the option to publish the peer review history of their article (what does this mean?). If published, this will include your full peer review and any attached files.

Reviewer #1: **Yes: **Xiangqun Ju

Reviewer #2: **Yes: **Dr Gadde Praveen

---

## [Author Response · Author response to Decision Letter 0]

23 Sep 2020

Editor: 

Although manuscript is suitable for publication but it is poorly written. Authors must improve the presentation of the paper in line with reviewers comments.

R: We respect your editorial comments but we do not agree that the manuscript was poorly written. We agree, however, that some aspects could be clarified, and the text improved, by considering the inputs from reviewers. We have made efforts to edit the text according to all suggestions from the reviewers. It is good to have an external view and clarify our article, but we respectfully disagree the report was poor. We want to highlight that the paper addresses the impact of COVID-19 to Brazil, which sadly is still the pandemic epicenter by September 2020. This means that the scope of the article was kept to Brazil but, in accordance with suggestions placed by the reviewers, the pandemic scenario and challenges imposed to the dental sector in other countries were addressed.

Reviewer #1: 

1. The sampling methodology is poorly described, for example, how do you know that a dentists (anonymously) belong to each state?

R: We disagree that methods were poorly described in the original paper. However, we have made efforts to improve clarity in all aspects raised by both reviewers. Information about the place of work of dentists, for instance, was available in the questionnaire submitted as a supplemental material. The instrument was a self-administered instrument, all responses were self-reported as detailed in the original manuscript. The participants were not asked to identify themselves but demographic data were collected and reported accordingly. In the original submission, the SURGE reporting guidance (Grimshaw 2014) was used for addressing important reporting aspects in the paper. Notwithstanding, we have revised the manuscript and included further details, now also addressing the CHERRIES reporting guideline (Eysenbach 2004) in order to cover all items that are considered relevant in online survey research. In our opinion, the sampling is adequately described, with enough details for one to understand how participants were recruited and responded to the questions presented in the survey. We have also recently published a methodological preprint manuscript (Moraes et al., 2020) addressing underlying data related to the methods used in this study. This information was addressed in the revised article and the study quoted in the references list.

Grimshaw J. SURGE (The SUrvey Reporting GuidelinE). In: Moher D, Altman DG, Schulz KF, Simera I, Wager E, editors. Guidelines for reporting health research: A user’s manual. 1st ed. Hoboken (NJ): John Wiley & Sons 2014; 206–213.

Eysenbach G. Improving the quality of Web surveys: The Checklist for Reporting Results of Internet E-Surveys (CHERRIES). J Med Internet Res 2004; 6(3):e34.

Moraes RR, Correa MB, Daneris A, Queiroz AB, Lopes JP, Lima GS, Cenci MS, D'Avila OP, Pannuti CM, Pereira-Cenci T, Demarco FF. Email vs. Instagram recruitment strategies for online survey research. medRxiv 2020.09.01.20186262; doi: https://doi.org/10.1101/2020.09.01.20186262

2. Covariates criteria and categories are simple and unclear, such as: did both the 1000 cases /100 deaths per million inhabitants in each state represent each state or a unit?

R: The units of analysis of both variables related to 1000 cases/100 deaths per million inhabitants are the states. This was the reason for the adoption of multilevel models, i.e., to consider in the analysis individuals nested in the same cluster (states). This information was clarified in the manuscript in the following passage:

“For analysis purposes, data were converted into thousands of cases and hundreds of deaths per one million inhabitants in each state. The units of analysis of both variables are the states. Multilevel mixed effect models were used to test the association between the contextual variables related to the status of the pandemic in each state and dentistry-related outcomes. Outcomes included decrease in number of patients assisted weekly (numerical), fear of contracting COVID-19 at work (no/a little vs. yes/a lot), and current work status (normal/reduced vs. not working/emergencies only). Linear and logistic models were used for numeric and binary outcomes. The models considered two levels of organization: dentist (level 1) and state (level 2). β-coefficients and Odds Ratios (OR) were reported. Contextual level variance was assessed using intraclass correlation coefficient (ICC) for linear models and Median Odds Ratio for logistic models (α=0.05).” 

3. Data analysis are also simple and unclear, especially for multilevel models: what was ‘null model’? and what is difference ‘OR’ and “MOR’? were models adjusted for any covariates?

R: The analyses were not adjusted by covariates because we have not hypothesized that individual factors would confound the association between the pandemic status in states and outcome variables. The Median Odds Ratio (MOR) is a measure of variance that can be attributed to a contextual level. It is analogous to the Intraclass Correlation Coefficient (ICC, also reported in the paper), being applied to binary outcomes. As stated by Merlo et al. (2006), “The aim of the median odds ratio (MOR) is to translate the area level variance in the widely used odds ratio (OR) scale, which has a consistent and intuitive interpretation. The MOR is defined as the median value of the odds ratio between the area at highest risk and the area at lowest risk when randomly picking out two areas the MOR can be conceptualized as the increased risk that (in median) would have if moving to another area with a higher risk.” In contrast, odds ratio (OR) was used as an effect measure in the logistic regression. Interpretation of each MOR and OR were provided in the results section. We do not believe that describing the explanation on what is the MOR in the manuscript is needed, but we do believe that the statistical models used, although may not be familiar to every researcher, were helpful in understanding the contextual effects on dentists-related outcomes. Results of contextual level variance of null models were added to the tables 4 and 5 according to your suggestion. Contextual level variances were estimated using ICC for linear models and MOR for logistic models. Both ICC and MOR were estimated for null (empty) and adjusted models, the tables now show both models.

Merlo J, Chaix B, Ohlsson H, et al. A brief conceptual tutorial of multilevel analysis in social epidemiology: using measures of clustering in multilevel logistic regression to investigate contextual phenomena. J Epidemiol Community Health. 2006;60(4):290-297. 

 

4. There is insufficient discussion of the relevance and the aim of the study following the findings/results.

5. Discussion section: it is short of strength and limitation reporting.

R: The discussion section was improved to cover more aspects and implications related to our results. In addition, the revised Discussion section also compares our findings to those of studies conducted in other countries, as suggested by Reviewer #2. Discussion on relevance of findings was also improved, as well as on strengths and limitations of the study.

Reviewer #2: 

1. Introduction doesn't cover enough background of research conducted so far on the topic.

R: The Intro section was revised to cover more research on the topic carried out in other countries. However, we want to highlight that the present study is a report on the challenges of COVID-19 pandemic imposed to the dental sector in Brazil, so the focus was kept to Brazil which, unfortunately, is still the pandemic epicenter as of September, 2020. Findings from other countries also were addressed in the Discussion section.

2. Methods section is too lengthy and has many repeated explanations

R: We understand your point but this comment is actually contrary to a comment placed by Reviewer #1, who indicates the methods were poorly described. We have made efforts to reduce the duplicity of information as much as possible. The methodology was described in full details to allow the reader to understand how the questionnaire was constructed and pre-tested, and how participants were recruited. This may also increase reproducibility of the methods, which is a topic that has been debated with emphasizes in science nowadays (Baker 2016). May we highlight that the methods section now includes two different reporting guidelines in order to cover all important aspects that are considered relevant in survey research: SURGE (Grimshaw 2014) and CHERRIES (Eysenbach 2004).

Baker M. 1,500 scientists lift the lid on reproducibility–Survey sheds light on the ‘crisis’ rocking research. Nature 533, 452–454.

Grimshaw J. SURGE (The SUrvey Reporting GuidelinE). In: Moher D, Altman DG, Schulz KF, Simera I, Wager E, editors. Guidelines for reporting health research: A user’s manual. 1st ed. Hoboken (NJ): John Wiley & Sons 2014; 206–213.

Eysenbach G. Improving the quality of Web surveys: The Checklist for Reporting Results of Internet E-Surveys (CHERRIES). J Med Internet Res 2004; 6(3):e34.

3. The response rate is very low though the sample size is adequate. since its a national wide survey, the response rate is critical. so i would suggest to improve the response rate and subsequent analysis

R: The response rates cannot be improved since the survey was conducted in May 2020. The issue was discussed in the text, and the fact that the study was conducted when Brazil was recently a new pandemic epicenter was highlighted. The pandemic scenario changes quickly and new responses collected now could not be considered belonging to the same context, which was as important aspect in the study, as discussed in a comment for Reviewer #1.

4. Discussion is too general and not based on important findings of the study. I would suggest to compare findings with other countries. It is more important to discuss on how these findings helps for change in policy as mentioned in the manuscript. further discuss the impact of pandemic on dentistry in other countries.

R: The discussion section was improved to cover more aspects related to our results compared to studies conducted in other countries, following your suggestion. Discussion on relevance of findings was also improved, as well as on strengths and limitations of the study.

5. Please check the references as some are repeated in the list.

R: We found one reference (Morita et al.) repeated in the list, thanks for noticing. The mistake was corrected. New references were quoted in the revised manuscript.

---

## [Decision Letter · Decision Letter 1]

30 Oct 2020

COVID-19 challenges to dentistry in the new pandemic epicenter: Brazil

PONE-D-20-21705R1

Dear Dr. Moraes,

We’re pleased to inform you that your manuscript has been judged scientifically suitable for publication and will be formally accepted for publication once it meets all outstanding technical requirements.

Kind regards,

Srinivas Goli, Ph.D.

Academic Editor

PLOS ONE

Additional Editor Comments (optional):

Revisions approved by the reviewers and also satisfactory to me. 

Reviewers' comments:

Reviewer's Responses to Questions

**Comments to the Author**

1. If the authors have adequately addressed your comments raised in a previous round of review and you feel that this manuscript is now acceptable for publication, you may indicate that here to bypass the “Comments to the Author” section, enter your conflict of interest statement in the “Confidential to Editor” section, and submit your "Accept" recommendation.

Reviewer #1: All comments have been addressed

Reviewer #2: All comments have been addressed

2. Is the manuscript technically sound, and do the data support the conclusions?

Reviewer #1: Yes

Reviewer #2: Yes

3. Has the statistical analysis been performed appropriately and rigorously? 

Reviewer #1: Yes

Reviewer #2: Yes

4. Have the authors made all data underlying the findings in their manuscript fully available?

Reviewer #1: Yes

Reviewer #2: Yes

5. Is the manuscript presented in an intelligible fashion and written in standard English?

Reviewer #1: Yes

Reviewer #2: Yes

6. Review Comments to the Author

Reviewer #1: The answers to my questions were well addressed, and the manuscript has been improved and valuable. I have two comments to be considered as minor corrections:

1. Please add full name for ‘SURGE’ and ‘CHERRIES’ (Line 111, page 5)

2. What variables were adjusted for? Please introduce/explain adjusted models in both the Methods section (data analysis) and Table 4 and 5 (footnotes).

Reviewer #2: The article adheres to appropriate reporting guidelines and community standards for data availability

7. PLOS authors have the option to publish the peer review history of their article (what does this mean?). If published, this will include your full peer review and any attached files.

Reviewer #1: **Yes: **Xiangqun Ju

Reviewer #2: **Yes: **Dr. Gadde Praveen

---

## [Editor Report · Acceptance letter]

17 Nov 2020

PONE-D-20-21705R1 

COVID-19 challenges to dentistry in the new pandemic epicenter: Brazil 

Dear Dr. Moraes:

I'm pleased to inform you that your manuscript has been deemed suitable for publication in PLOS ONE. Congratulations! Your manuscript is now with our production department. 

Kind regards, 

on behalf of

Dr. Srinivas Goli 

Academic Editor

PLOS ONE